# Spatial variation and factors associated with home delivery after ANC visit in Ethiopia; spatial and multilevel analysis

Hiwotie Getaneh Ayalew[1]*, Alemneh Mekuriaw Liyew[2], Zemenu Tadesse Tessema[2], Misganaw Gebrie Worku[3], Getayeneh Antehunegn Tesema[2], Tesfa Sewunet Alamneh[2], Achamyeleh Birhanu Teshale[2], Yigizie Yeshaw[2,4], Adugnaw Zeleke Alem[2]

1 Department of Midwifery, School of Nursing and Midwifery, College of Medicine and Health Sciences, Wollo University, Dessie, Ethiopia, 2 Department of Epidemiology and Biostatistics, Institute of Public Health, College of Medicine and Health Sciences and Comprehensive Specialized Hospital, University of Gondar, Gondar, Ethiopia, 3 Department of Human Anatomy, College of Medicine and Health Sciences and Comprehensive Specialized Hospital, University of Gondar, Gondar, Ethiopia, 4 Department of Human Physiology, College of Medicine and Health Sciences and Comprehensive Specialized Hospital, University of Gondar, Gondar, Ethiopia

* hiwotiegeta27@gmail.com

**Data Availability Statement:** One can get data at the link: www.measuredhs.com after being authorized user of demographic and health survey data.

## Abstract

### Introduction

Institutional delivery is crucial to reduce maternal and neonatal mortality as well as serious morbidities. However, in Ethiopia, home delivery (attended by an unskilled birth attendant) after antenatal care (ANC) visit is highly in practice. Therefore, this study aimed to assess the spatial variation and determinants of home delivery after antenatal care visits in Ethiopia.

### Method

A secondary data analysis was conducted using the 2019 mini Ethiopian demographic and health survey. A total of 2,923 women who had ANC visits were included. Spatial analysis was done by using GIS 10.7 and SaTscan 9.6. The risk areas for home delivery from GIS and spatial scan statistics results were reported. A multi-level logistic regression model was fitted using Stata14 to identify individual and community-level factors associated with home delivery after ANC visit. Finally, AOR with 95% CI and random effects were reported.

### Result

Home delivery after ANC visit was spatially clustered in Ethiopia(Moran's index = 0.52, p-value <0.01). The primary clusters were detected in Oromia and SNNP region (LLR = 37.48, p < 0.001 and RR = 2.30) and secondary clusters were located in Benishangul Gumuz, Amhara, Tigray and Afar (LLR = 29.45, p<0.001 and RR = 1.54). Being rural resident (AOR = 2.52; 95%CI 1.09–5.78), having no formal education (AOR = 3.19;95% CI 1.11–9.16), being in the poor (AOR = 2.20;95%CI 1.51–3.22) and middle wealth index (AOR = 2.07;95% CI 1.44–2.98), having one ANC visit (AOR = 2.64; 95% CI 1.41–4.94), and living

**Funding:** The authors received no specific funding for this work.

**Competing interests:** The authors have declared that no competing interests exist.

**Abbreviations:** ANC, Antenatal Care; AOR, Adjusted odds Ratio; EDHS, Ethiopian Demographic and Health Survey; ICC, Intra-cluster Correlation Coefficient; LLR, log-likelihood Ratio; MEDHS, Mini Ethiopian Demographic and Health Survey; PCV, Proportional Change in Variance; RR, Relative Risk; SNNP, South Nation Nationality and Peoples Region.

in the agrarian region (AOR = 3.63; 95%CI 1.03–12.77) had increased the odds of home delivery after ANC visit.

## Conclusion and recommendation

Home delivery after ANC visit was spatially clustered in Ethiopia. Factors like maternal education, wealth index, number of ANC visits, residency and region were significantly associated with home delivery after ANC visit. Therefore, it is better to increase the number of ANC contact by giving health education, especially for women with low levels of education and better to improve the wealth status of women. A special strategy is also vital to reduce home delivery after ANC visit in those high-risk regions.

## Introduction

Home delivery is childbearing in women's or other homes without unskilled health personnel and a non-equipped clinical setting. In Africa, home delivery is high and institutional deliveries attended by non-trained individuals are highly in practice [1]. In those countries, maternal mortalities are due to direct obstetric causes including postpartum hemorrhage, obstructed labor, pregnancy-induced hypertension and sepsis [2]. Maternal morbidity and mortality is a global health challenge and developing countries contributed to about 97% of maternal death [3]. It is also estimated that 810 women across the world die each day as a result of pregnancy and childbirth-related problems and a huge number of these deaths occur in developing countries. In Sub-Saharan Africa, childbearing women face a 1 in 39 risks of dying in childbirth [3, 4]. In Ethiopia, most maternal deaths are related to pregnancy and childbirth and the current maternal mortality ratio is 412 per 100,000 live births [5]. Place of delivery is crucial to reduce maternal morbidity and mortality as well as to reduce serious neonatal illness but in Ethiopia, home delivery attended by an unskilled birth attendant is highly practiced (4,6). However, a large number of these maternal deaths could be prevented through the presence and utilization of institutional delivery and skilled health care provider at delivery services. Currently, in Ethiopia, despite ANC coverage(74%), home delivery attended by unskilled birth attendants was 50% [6].

Even if, an ANC visit is a key factor for institutional delivery, many women who had ANC follow up, gave birth at home usually without skilled birth attendants [7, 8]. Mothers who had ANC visits also give birth at home [9]. In Ethiopia, the 2019 MEDHS report shows that mothers who had ANC visits didn't give birth at health institutions, since ANC and institutional delivery coverage are 74% and 50% respectively [5].

Different studies identified factors that were significantly associated with home delivery after ANC visit. These were residence, inadequate knowledge of pregnancy-related problems, educational status of the mother (6, 8, 10), women's income, number of ANC visits, parity, ANC follow up and distance from a health facility (6, 8, 9, 10, 11).

Home delivery after ANC visit is a major public health problem in Ethiopia, although limited studies have been conducted on the determinant factors and the spatial clustering of home delivery just after ANC visit. Therefore, this study aims to assess the spatial variation and determinants of home delivery just after ANC visits across regions in Ethiopia. Besides, the findings from the geostatistical analysis are helpful to locate the high-risk areas of home delivery after ANC visits which coud in turn has a policy implication for strategic intervention and allocation of resources [10].

## Method

### Data source, study population and sampling technique

Secondary data analysis was done based on the mini Ethiopian demographic and health survey 2019 dataset, which was a crosssectional survey conducted from March 21, 2019, to June 28, 2019. The source population was all pregnant women who had ANC visits within five years preceding the survey in Ethiopia. Whereas, the study population was all pregnant women aged 15–49 years who were pregnant or had last birth and who had ANC visits in the selected households five years preceding the survey. A two-stage stratified cluster sampling was used in the 2019 mini Ethiopian Demographic and Health survey by taking enumeration areas as primary sampling units and households as secondary sampling units. The sample included 305 enumeration areas, 93 in urban and 212 in rural areas. A total of 8,885 reproductive-age women were interviewed in the 2019 survey. For this study, participants with missing in the outcome variable were excluded. Therefore a total of 2,923(weighted) pregnant women who had ANC visits five years preceding the survey were included in the analysis [5].

### Study variables

**Dependent variable.** The outcome variable of this study was home delivery after ANC visit, which was defined as mothers who gave birth at her or another's home after ANC visits by unskilled birth attendants. For analysis, it was coded as "1" for women who had home delivery and coded as "0" for women who had institutional delivery.

**Independent variables.** In this study, both individual and community-level factors were considered. Individual-level factors included were the educational status of the mother (no formal education, primary, secondary, higher), marital status (married, unmarried), maternal age (15–24, 25–34 and 35–49 years), wealth status (poor, middle, rich), history contraceptive utilization (no, yes), media exposure (no, yes), the number of ANC follow-up (one visit, 2–4 visit, above 4 visits), time of ANC visit (first trimester, second trimester, third trimester), birth interval (short, long), whereas the community-level factors included were the place of residency (urban, rural), region,community-level poverty and community media exposure. The region was categorized into three as agrarian (Tigray, Amhara, Oromia, and Sothern Nations Nationalities and Peoples Region), pastoral (Afar, Somali, Benishangul, and Gambella), and urban (Harari, Dire Dawa and Addis Ababa). The community-level media exposure was obtained by aggregating the individual-level media exposure in each cluster by using the proportion of those who had media exposure and this community-level media exposure shows the overall media exposure in the community. Median values were used to categorize as high and low because the aggregated variable had a skewed distribution. Community-level poverty level was also obtained by an aggregated proportion of poor women which shows the overall poverty status within the cluster. It was categorized into two categories based on the median value a higher proportion of poor mothers and a lower proportion of mothers within a cluster.

### Data management and analysis

The data was accessed from the DHS program's official database after permission was granted through an online request [11]. The outcome variable with important predictors was extracted from MEDHS 2019, maternal individual records data set. Labeling, recoding and analysis were done using STATA 14. The data were weighted using sampling weight, primary sampling unit, and strata before any statistical analysis to restore the representativeness of the survey for unequal sample sizes across clusters and to tell the STATA to take into account the sampling design when calculating standard errors to get reliable statistical estimates.

**Spatial analysis.** The weighted frequency of the outcome variable with cluster number and geographic coordinate data was used. A total of 305 clusters having longitude and latitude were included in the spatial analysis. Spatial autocorrelation analysis (Global Moran's I index) and hotspot analysis were conducted by using ArcGIS 10.7, while spatial scan statistics was done by SaTscan 9.6. Finally, a weighted proportion of home delivery after ANC visit per cluster was used for spatial autocorrelation analysis and hotspot analysis. For spatial scan statistics number of women who had home delivery after ANC visit and the number of women who had institutional delivery were considered as cases and controls respectively to fit the Bernoulli model [12].

**Spatial autocorrelation analysis.** The Global Moran's I statistic test was used to measure whether home delivery after ANC visit patterns were randomly distributed, dispersed, or clustered in Ethiopia. The calculated Moran's I values close to −1 indicate home delivery was dispersed, whereas Global Moran's I close to +1 indicate home delivery after ANC visit was clustered and if the Global Moran's I value zero home delivery were distributed randomly [13, 14].

**Hot spot (Getis-Ord Gi$^*$) analysis.** The nature of the variation of spatial autocorrelation over the study setting was investigated by calculating Gi$^*$ statistic, for each area by using Gettis-Ord Gi$^*$ statistics. Z-score is computed to determine the statistical significance of clustering, and the p-value is computed for the significance [14]. If the z-score is between −1.96 and +1.96, the p-value would be larger than 0.05, and could not reject the null hypothesis; the pattern exhibited could very likely be due to random spatial processes. If the z-score falls outside the range, the observed spatial pattern is probably unlikely to be because of random chance, and the p-value would be small. In this condition, the null hypothesis is rejected and dealing with what might be causing the statistically significant spatial pattern in the data is considered. Statistical output with high Gi$^*$ indicates a "hotspot" whereas low Gi$^*$ indicates a "cold spot" [13, 15].

**Spatial scan statistical analysis.** Spatial scan statistical analysis or the Bernoulli-based model was employed to test for the presence of statistically significant spatial clusters of home delivery after ANC visit using Kuldorff's SaTScan version 9.6software. The spatial Scan statistical method is commonly recommended that it is better than others in detecting local clusters and has higher power as compared to available spatial statistical methods [16]. The presence of statistically significant clusters of home delivery after ANC visit was tested by using spatial scan statistical analysis. It uses a scanning window that moves across the study area [15, 17]. Women who had home delivery after ANC visit were taken as cases and those who had institution delivery were taken as controls to fit the Bernoulli model. The number of cases in each location had a Bernoulli distribution and the model requires data with or without a disease. For each potential cluster, a likelihood ratio test statistic was used to determine the number of observed home delivery cases within the potential cluster was significantly higher than expected or not. The primary and secondary clusters were identified and assigned p-values and ranked based on their likelihood ratio test, based on 999 Monte Carlo replications [18, 19].

**Multi-level logistic regression analysis.** The outcome variable was binary; home delivery and institution delivery. Because of the hierarchical nature of data and the binary outcome variable, the multi-level logistic regression model was used. Before fitting the multilevel logistic regression model for individual and community level variables the chi-square assumption was checked. In the Bivariable multilevel logistic regression model and Variables which were statistically significant at a p-value less than 0.2 in the Bivariable multilevel logistic regression analysis were considered for model adjustments for the multivariable multilevel logistic regression model. Four models were fitted. The first was the null model containing no exposure variables

which was used to check variation in the community and provide evidence to assess random effects. The second model was the multivariable model adjustment for individual-level variables and model three was adjusted for community-level factors. In the fourth model, variables from both individual and community-level variables were fitted. Regarding the measures of variation (random effects) intracluster correlation coefficient (ICC) and Proportional Change in Community Variance (PCV) were used.

The ICC (Intracluster correlation coefficient) quantifies the degree of heterogeneity of home delivery after ANC visit between clusters (enumeration areas).

$$ICC = 62/(62 + \pi 2/3)$$

Where:—$6^2$-between cluster variance, $\pi^2/3$ -within-cluster variance

Whereas PCV measures the total variation attributed to individual-level factors and community-level factors in the multilevel model as compared to the null model which was computed by using the following formula;PCV = (variance in (null model)-variance in (full model))/ variance (null model) [20].

Since the models were hierarchical and nested, the model comparison was done using deviance and log likely hood ratio. Consequently, the best-fitted model for the data was used. Finally, the adjusted odds ratio with a 95% confidence interval was reported for statistically significant variables.

## Ethical consideration

This study was a secondary analysis of the 2019 mini Ethiopian Demographic and Health Survey data.For conducting this study; we registered and requested the dataset from DHS online archive. The approval letter for the use of the data set was also gained from the Measure DHS. No information obtained from the data set was disclosed to any third person.

## Result

### Individual and community-level characteristics of the respondents

Among a total of 2,923 participants, the mean age of the participants was 28 years (SD±6.42), and 1,283 (43.87%) participants had no formal education. Most of the respondents 2,754 (94.22%) were married. Regarding media, 1,664 (56.94%) participants had no media exposure. A total of 1348 (46.12%) participants were rich and most of the respondents 2,581 (88.30%) had a history of contraceptive use. Looking at the timing of ANC follow up above half of the participants 1,563 (53.48%) were started in the second trimester and 2,014(68.91%) participants had 2–4 ANC visits.

A total of 305 communities (clusters) were included in the study. about 89% of study participants lived in agrarian regions. Around two-thirds (60.14%) of the participants lived in communities with low poverty levels and nearly half of them (51.31%) lived in communities with high community media exposure (Table 1).

### Spatial analysis results

**Spatial autocorrelation of home delivery after ANC visit in Ethiopia.**   This study identified thathome delivery was clustered in Ethiopia with Global Moran's I = 0.52 and p-value = 0.000.A Z-score of 11.34 indicated that there is less than a 1% likelihood that this clustered pattern could be the result of random chance. The bright red and blue colors to the end tails indicate an increased significance level (Fig 1).

**Table 1. Individual and community-level characteristics of study participants who had home delivery after ANC visit in Ethiopia, MEDHS 2019 (N = 2,923).**

| Individual level variables | Frequency (weighted) | Percentage |
|---|---|---|
| Maternal age (years) | | |
| 15–24 | 759 | 25.95 |
| 25–34 | 1,550 | 53.03 |
| 35–49 | 615 | 21.02 |
| Maternal education | | |
| No formal education | 1,283 | 43.87 |
| Primary education | 1,153 | 39.46 |
| Secondary education | 335 | 11.45 |
| Higher education | 153 | 5.22 |
| Marital status | | |
| Married | 2,754 | 94.22 |
| Unmarried | 169 | 5.78 |
| Wealth index | | |
| Poor | 986 | 33.74 |
| Middle | 586 | 20.13 |
| Rich | 1,348 | 46.12 |
| Media exposure | | |
| No | 1,664 | 56.94 |
| Yes | 1,259 | 43.06 |
| Parity | | |
| Primiparous | 689 | 23.57 |
| Multiparous | 1,662 | 56.87 |
| Grand multiparous | 572 | 19.57 |
| Sex of household head | | |
| Male | 2,538 | 86.82 |
| Female | 385 | 13.18 |
| Previous contraceptive use | | |
| No | 342 | 11.70 |
| Yes | 2,581 | 88.30 |
| Birth interval | | |
| Short | 419 | 18.90 |
| Long | 2,504 | 81.10 |
| Time of ANC visit | | |
| First trimester | 1,092 | 37.37 |
| Secondtrimester | 1,563 | 53.48 |
| Third trimester | 268 | 9.16 |
| Number of ANC visit | | |
| 1 visit | 130 | 4.46 |
| 2–4 visit | 2,014 | 68.91 |
| Above 4 visit | 779 | 26.63 |
| **Community-level variables** | | |
| Place of residency | | |
| Urban | 871 | 29.81 |
| Rural | 2,052 | 70.19 |
| Region | | |
| Urban | 150 | 5.11 |

*(Continued)*

**Table 1.** (Continued)

| Individual level variables | Frequency (weighted) | Percentage |
|---|---|---|
| Agrarian | 2,620 | 89.61 |
| Pastoral | 154 | 5.28 |
| Community poverty | | |
| Low | 1,758 | 60.14 |
| High | 1,165 | 39.86 |
| Community media exposure | | |
| Low | 1,423 | 48.69 |
| High | 1,500 | 51.31 |

**Hot spot analysis of home delivery after ANC visit in Ethiopia.** Hot spot analysis was performed to identify high-risk areas for having home delivery after ANC visitin Ethiopia. The green color indicates less risky areas for home delivery in Ethiopia. The red color indicates significant high-risk areas that were detected in the Somalia,Amhara, Afar and Oromia and

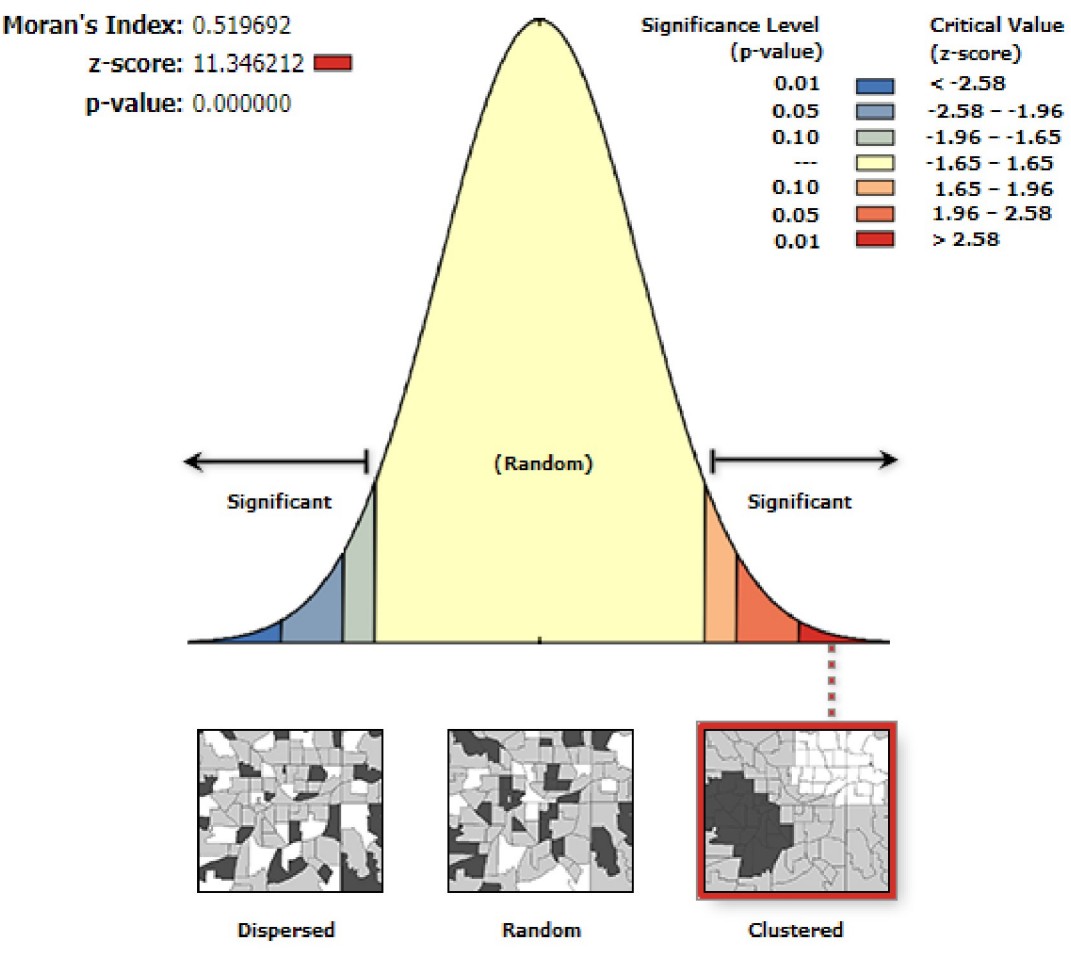

Given the z-score of 11.3462124419, there is a less than 1% likelihood that this clustered pattern could be the result of random chance.

**Fig 1. Spatial autocorrelation of home delivery in Ethiopia, MEDHS 2019.**

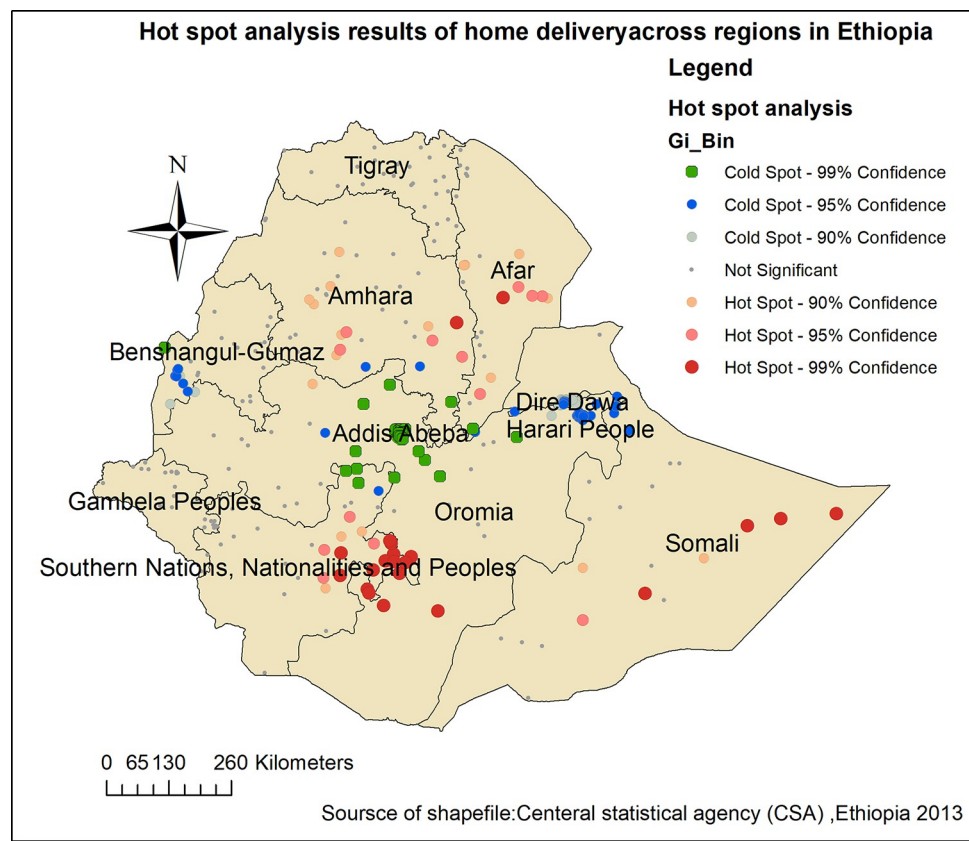

**Fig 2. Hot spot analysis of home delivery after ANC visit in Ethiopia, MEDHS 2019.**

SNNP regions. However, cold spot areas were detected in Dire Dawa, Addis Abeba and Harari peoples regions which, were indicated by green colored dots (Fig 2).

**Spatial SaTScan analysis of home delivery after ANC visit in Ethiopia (Bernoulli model).** Spatial scan statistics identified 133 significant most likely clusters, of which 14 primary, 119 secondary clusters were identified.The primary clusters' spatial window was located in southwest Oromia and SNNP regions, which was centered at (6.362562 N, 38.759281 E) with a 111.22 kmradius, and Log-Likelihood ratio (LLR) of 37.48, at p < 0.001. This showed that women within the primary scanning window had 2.30 times higher risk of home delivery after ANC visit as compared to women outside the scanning window. The secondary clusters scanning window was located in the north part of Benishangul Gumuz, Amhara, Tigray and Afar regions, which was centered at (14.154958 N, 39.329700 E) with a 553.25 km radius, and LLR of 29.45 at p<0.001. This showed that women who live within the secondary scanning window had a 1.54 times higher risk of home delivery after ANC visit as compared to women outside the scanning window. In addition, an other secondary clusters' scanning window was located in Somalia and some parts of the Oromia regions. It was centered at (6.639662 N, 44.465855 E) with a 390.28 km radius, with LLR 16.27 at a p-value<0.01. This showed that women who live within the tertiary scanning window had a 1.88 times higher risk of home delivery after ANC visit as compared to women outside the scanning window (Fig 3, Table 2).

### Random effect analysis results

Variance component analysis was performed to decompose the total variance of home delivery after ANC visit between clusters in the null model. The cluster-level variance which indicates

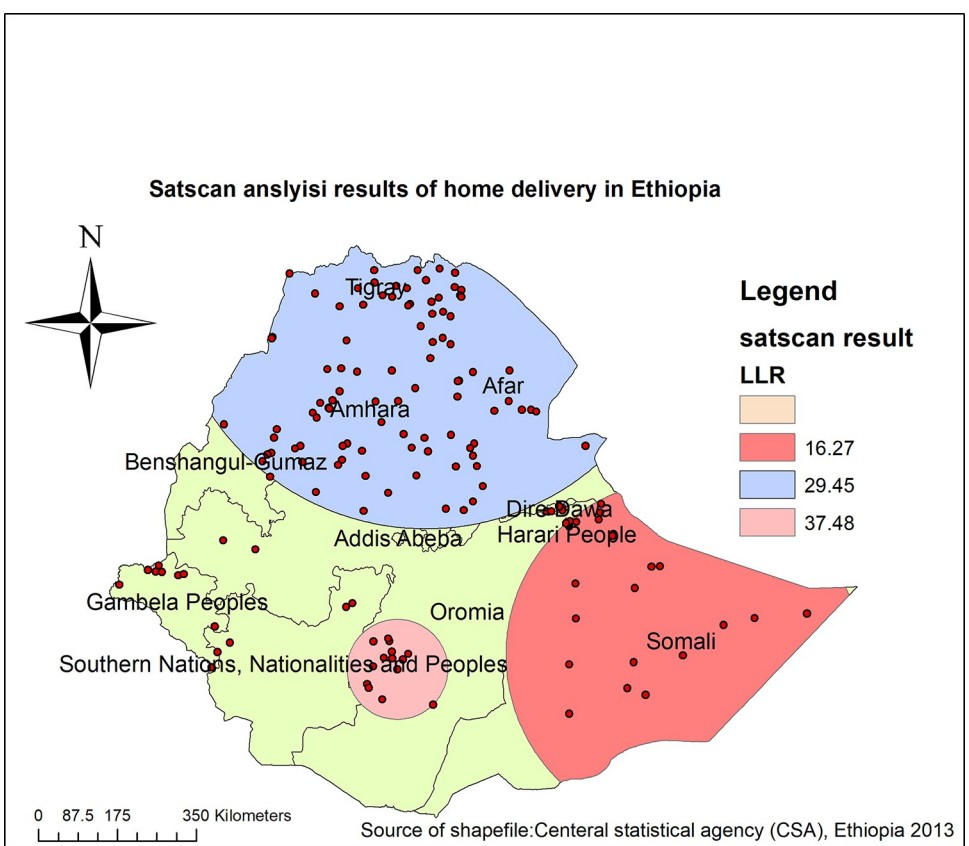

**Fig 3. Spatial scan analysis home delivery after ANC visit in Ethiopia, MEDHS 2019.**

the total variance of home delivery that can be attributed to the context of the community in which the mothers were living was estimated. The applicability of the multi-level logistic regression model in the analysis was justified by the significance of the community-level variance [community variance = 3.05; standard error (SE) = 0.46; P-value = 0.00], indicating the existence of significant differences between communities regarding home delivery after ANC visit incidence. The community variance was expressed as an intracluster correlation coefficient (ICC). The ICC was 0.4811 which revealed that 48.11% of the total variance of home delivery after ANC visitin Ethiopia could be attributed to the context of the communities

**Table 2. Most likely clusters of spatial scan statistics analysis of home delivery after ANC visit in Ethiopia, MEDHS 2019.**

| Most likely clusters | Enumeration areas(clusters) identified | Coordinate/radius | Population | Case | RR | LLR | p-value |
|---|---|---|---|---|---|---|---|
| First most likely cluster | 113, 183, 186, 182, 181, 117, 115, 185, 187, 172, 89, 188, 184, 114 | (6.362562N, 38.759281) / 111.22 km | 128 | 86 | 2.30 | 37.48 | <0.01 |
| Second most likely cluster | 16, 10, 15, 11, 3, 17, 35, 14, 39, 12, 2, 5, 37, 25, 27, 13, 38, 36,7, 23, 1, 6, 24, 20, 9, 8, 19, 18, 22, 78, 56, 45, 29, 46, 21, 62,83, 44, 82, 34, 61, 58, 84, 4, 57, 30, 33, 60, 59, 65, 64, 63, 54,81, 31, 55, 74, 85, 26, 51, 47, 32, 66, 75, 48, 53, 70, 49, 71, 76, 68, 50, 67, 72, 52, 73, 79, 165, 80, 43, 100, 162, 77, 40, 126, 69,163, 42, 148, 119, 99, 166, 159, 164 | (14.154958 N,39.32970E) / 553.25 km | 981 | 384 | 1.54 | 29.45 | <0.01 |
| Third most likely cluster | 135, 123, 140, 137, 138, 124, 131, 132, 122, 134, 136, 142, 133, 139,129, 121, 130, 107, 250, 128, 248, 249, 255, 254, 247 | (6.639662 N,44.465855E) / 390.28 km | 114 | 64 | 1.88 | 16.27 | <0.01 |

**Table 3. Random effects and model fitness.**

| Random effects | Model l | Model ll | Model lll | Model lV |
|---|---|---|---|---|
| Community variance(SE) | 3.05(0.46) | 2.08(0.36) | 2.09(0.32) | 2.06(0.31) |
| ICC (%) | 48.11(0.40–0.55) | 38.79(0.30–0 .47) | 38.89(0.31–0.46) | 38.60(0.31–0.45) |
| PCV (%) | Reference | 31.8 | 31.47 | 32.45 |
| **Model fitness** | **Model I** | **Model ll** | **Model lll** | **Model lV** |
| Log likelihood | -1513.41 | -1125.72 | -1462.03 | -1107.13 |
| Deviance(-2LLR) | 3026.82 | 2251.77 | 2924.06 | 2214.26 |

ICC: intracluster correlation coefficient; PCV: proportional change in variance; SE: standard error; LLR: log-likelihood ratio.

(clusters) where the mothers were living. The PCV in this model was 32.45%. It suggested that 32.45% of community variance observed in the null model was explained by both community and individual-level variables.

## Model comparison

A model with a high likelihood (-1107.13) and low (2214.26) deviance was the best model. The combined multilevel logistic regression model (model IV) was thebest-fitted model in this study (Table 3).

## Fixed effect analysis results

In bi-variable multilevel logistic regression analysis wealth index, maternal age, marital status, media exposure, maternal education, sex of household head, history of contraceptive use, birth interval,number of ANC visits,time of ANC visit, place of residence, region,community poverty level and community media exposure were significant at p-value<0.2 and fitted for Multivariable multilevel analysis. Multivariable multilevel logistic regression analysis was fitted to identify factors associated with home delivery after ANC visit. Thus, in the final model (Model IV) Maternal Education, wealth index, number of ANC visits; residency and region were significantly associated with home deliveryafter ANC visit.

After keeping another individual, community-level factors and random effect constant the odds of having home delivery after ANC visit among women who had no formal education was 3.19 (AOR = 3.19;95% CI 1.11–9.16) times higher as compared to women who had higher education. Women with poor and middle wealth index had 2.20(AOR = 2.20; 95%CI 1.51–3.22) and 2.07(AOR = 2.07;95% CI 1.44–2.98) respectively times higher odds ofhome delivery after ANC visitas compared to women with rich wealth index. Women who had one ANC visit had 2.64(AOR = 2.64; 95% CI 1.41–4.94) times higher odds of home delivery after ANC visit as compared to women who had four and above ANC visits.The odds of having home delivery after ANC visit among rural women was 2.51(AOR = 2.52; 95%CI 1.09–5.78) times higher as compared to urban women. Women who had lived in the agrarian regions had 3.63 (AOR = 3.63; 95%CI 1.03–12.77) times higher odds of home delivery after ANC visit as compared to urban regions (Table 4).

## Discussion

In this study among the spatial analysis methods hotspot analysis and SaTScan statisticsanalysis were used to demonstrate the clustering of home delivery after ANC visits in Ethiopia. In hotspot analysis, hot spot areas of home delivery after ANC visit were found in Somalia, Amhara, Afar, Oromia and SNNP region. This might be due to sudden onset of labor, lack of

**Table 4. Multilevel logistic regression analysis of individual and community-level factors associated with home delivery after ANC visit in Ethiopia, MEDHS 2019.**

| Characteristics Fixed effect | Model I | Model II AOR(95%CI) | Model lll AOR(95%CI) | Model IV AOR(95%CI) |
|---|---|---|---|---|
| Maternal age (years) | | | | |
| 15–24 | - | 1.00 | - | 1.00 |
| 25–34 | - | 1.03(0.70–1.52) | - | 1.10(0.75–1.62) |
| 35–49 | - | 0.84(0.52–1.34) | - | 0.91(0.57–1.46) |
| Maternal education | | | | |
| No | - | 4.39(1.62–11.92) | - | 3.19(1.11–9.16)* |
| Primary | - | 2.78(1.03–7.49) | - | 2.08(.73–5.90) |
| Secondary | - | 1.42(0.49–4.12) | - | 1.10(0.36–3.38) |
| Higher | - | 1.00 | - | 1.00 |
| Marital status | | | | |
| Married | - | 1.00 | - | 1.00 |
| Unmarried | - | 1.26(0.64–2.48) | - | 1.35(0.67–2.69) |
| Wealth index | | | | |
| Poor | - | 2.91 (2.02–4.19) | - | 2.20(1.51–3.22)* |
| Middle | - | 2.58(1.80–3.69) | - | 2.07(1.44–2.98)* |
| Rich | - | 1.00 | - | 1.00 |
| Media exposure | | | | |
| No | - | 1.20 (0.89–1.61) | - | 1.12(0.83–1.51) |
| Yes | - | 1.00 | - | 1.00 |
| Sex of household head | | | | |
| Male | - | 1.00 | - | 1.00 |
| Female | - | 1.36 (0.85–2.16) | - | 1.48(0.92–2.40) |
| Previouscontraceptive utilization | | | | |
| No | - | 1.38(0.97–1.98) | - | 1.38(0.96–1.97) |
| Yes | - | 1.00 | - | |
| Time of ANC visit | | | | |
| First trimester | - | 1.00 | - | 1.00 |
| Second trimester | - | 1.28(0.97–1.69) | - | 1.24(0.93–1.64) |
| Third trimester | | 2.44(0.56–3.82) | | 2.33(0.49–3.66) |
| Number of ANC visit | - | | - | |
| One visit | - | 2.82(1.51–5.25) | - | 2.64(1.41–4.94)* |
| 2–4 visit | | 1.08(0.80–1.46) | | 1.04(0.77–1.41) |
| Above 4 visit | | 1.00 | | 1.00 |
| Birth interval | - | | | |
| Long | - | 0.85(0.63–1.15) | - | 0.84(0.63–1.14) |
| Short | - | 1.00 | - | 1.00 |
| Residency | | | | |
| Urban | - | - | 1.00 | 1.00 |
| Rural | - | - | 3.57(1.66–7.69) | 2.51(1.09–5.78)* |
| Region | | | | |
| Agrarian | - | - | 3.34(1.16–9.61) | 3.63(1.03–12.77)* |
| Pastoral | - | - | 2.97(0.92–9.53) | 2.08(0.52–8.27) |
| Urban | - | - | 1.00 | 1.00 |
| Community poverty | | | | |
| Low | - | - | 1.00 | 1.00 |
| High | - | - | 2.88(1.64–5.07) | 1.79(0.97–3.29) |
| Community media exposure | | | | |

*(Continued)*

**Table 4.** (Continued)

| Characteristics Fixed effect | Model I | Model II AOR(95%CI) | Model III AOR(95%CI) | Model IV AOR(95%CI) |
|---|---|---|---|---|
| Low | - | - | 1.92(1.06–3.48) | 1.57(0.84–2.94) |
| High | - | - | 1.00 | 1.00 |

ANC: antenatal care

*significant; 1.00: reference.

privacy, absence of transportation, long-distance and low accessibility of health service facilities resulting in-home delivery [9, 21, 22]. Another possible justification for this finding might be regions like Addis Ababa, Dire Dawa and Harari were urban areas where, most of the mothers had better education levels and could access media and health facilities compared to the other regions [23, 24]. Besides, maternal health care service-seeking behavior of women is low in those hotspot areas like Somalia and Afar which could lead to home birth [25–27].

Moreover, Spatial scan statistics identified 133 significant most likely clusters, of which 14 primary, 119 secondary clusters. These most likely clusters' were located in Oromia, Amhara, and Tigray, Afar, Somalia, Benishangul Gumuzand SNNP regions. This could be due to the inequality in the distribution of maternal health services and the inaccessibility of infrastructure [28, 29]. Besides, the mobile lifestyle of nomads, negative Attitude toward skilled birth attendants and low perception of institutional delivery could be potential resons that might reduce maternal health care service utilization [30, 31].

Maternal education was significantly associated with home delivery after ANC visit. Women with no formal education had more than three times higher odds of having home delivery after ANC visit as compared to women who had higher education. This finding was supported by studies done in Ethiopia [9, 32, 33], Kenya [34] and Tanzania [22, 35]. This mightbe due to that educated women had a better level of understanding about skilled delivery service utilization, healthcare-seeking behavior and maternal survival [36]. Another possible reason might be that educated women can easily be exposed to media and can easily access information about the health benefits of institutional delivery [37]. In addition to this most women who had higher education levels lived in urban areas, which help them to easily access health institutions for delivery care services [38].

The wealth status of women was another factor associated with home delivery after ANC visit. The odds of having home delivery after ANC visit was about two times higher for women within a community with the poor wealth status as compared to those from communities with rich wealth status.This finding was in line with studies done in Ethiopia [33, 39], Tanzania [22] and Nepal [40]. This might be due to the demand for different costs related to transportation and food that might not be afforded by the mother's in low level of income [31, 41]. Possibly it might be due to that mostly poor women live in rural areas in Ethiopia in the absence of transportation and long-distance travel and access to health facilities which may lead to home delivery [42–45].

The number of ANC visits was also significantly associated with home delivery after ANC visits. Women who had one ANC visit had about 64% increased odds of home delivery after ANC visit as compared to women who had four and above ANC visits. This finding was consistent with studies done in Ethiopia [2, 9, 32] and Nepal [40].The possible reason might be due to the increased number of ANC contact with health professionals enhancing the chance to advise about birth preparedness and place of delivery [46]. In addition,women who had fewer than four ANC visits didn't have a better number of contact with skilled health care

providers but, an increased number of ANC visits gave chance for the mothers to be aware of the importance of institutional delivery by skilled birth attendants.

The Place of residency was another factor which strongly associate with home delivery after ANC visit. The odds of having home delivery after ANC visit among rural women were more than two times higher as compared to urban women. This finding was supported by different studies done in Ethiopia [33, 39, 47, 48] and Kenya [49]. The possible explanation might be due to rural residents usually have little access to health facilities as compared to urban residents [33]. This may also be due to distance to the health facilities, ambulance delays due to distance and difficult roads, inaccessibility of information about the advantage of institutional delivery and low educational status of women from rural areas [50].

Finally, the region woman lived in was also a significant factor. Women who had lived in the agrarian regions had a 34% increased odds of home delivery as compared to the urban regions. This finding was concordant with studies done in Ethiopia [51, 52], Kenya [53], Ghana [54] and Nigeria [55]. The possible explanation might be due to scarcity of resources and poor clinical settings,in which most of the services are concentrated in urban areas or city administrates [56, 57]. In addition, those women who live in agrarian regions are farmers commonly had sudden labor onset and short duration labor due to hard work and exercise, which lead to home delivery [58, 59]. It might also be due to regions being attributable to the differences in access to health services, information and social and cultural attributes [55].

The strength of this study was it used a large data set and applied sampling weight to make it nationally representative and to give reliable estimates and also used an advanced model for analysis. Whereas, the limitation of this study could be the possibility of committing social desirability since it was a community-based national survey.

## Conclusion

Home delivery after ANC visit was spatially clustered in Ethiopia. Women from Somalia, Amhara, Afar, Oromia, Benishangul Gumuz,Gambella, Tigrayand SNNP regions had a higher risk of home delivery after ANC visit. Factors like maternal education, wealth index, number of ANC visits, residency and region were significantly associated with home delivery after ANC visit in Ethiopia. Therefore, it is better to increase the number of ANC contact by giving health education, particularly for women with low levels of education and better to improve women's income by setting different strategies and job opportunities at a national level to support women in those high-risk regions.

## Acknowledgments

We would like to greatly acknowledge MEASUREDHS for granting us to access the Demographic and Health Surveys data set.

## Author Contributions

**Conceptualization:** Hiwotie Getaneh Ayalew, Alemneh Mekuriaw Liyew, Zemenu Tadesse Tessema, Misganaw Gebrie Worku, Getayeneh Antehunegn Tesema, Tesfa Sewunet Alamneh, Achamyeleh Birhanu Teshale, Yigizie Yeshaw, Adugnaw Zeleke Alem.

**Formal analysis:** Hiwotie Getaneh Ayalew, Alemneh Mekuriaw Liyew, Zemenu Tadesse Tessema, Misganaw Gebrie Worku, Getayeneh Antehunegn Tesema, Tesfa Sewunet Alamneh, Achamyeleh Birhanu Teshale, Yigizie Yeshaw, Adugnaw Zeleke Alem.

**Investigation:** Hiwotie Getaneh Ayalew, Alemneh Mekuriaw Liyew, Zemenu Tadesse Tessema, Misganaw Gebrie Worku, Getayeneh Antehunegn Tesema, Tesfa Sewunet Alamneh, Achamyeleh Birhanu Teshale, Yigizie Yeshaw, Adugnaw Zeleke Alem.

**Methodology:** Hiwotie Getaneh Ayalew, Alemneh Mekuriaw Liyew, Zemenu Tadesse Tessema, Misganaw Gebrie Worku, Getayeneh Antehunegn Tesema, Tesfa Sewunet Alamneh, Achamyeleh Birhanu Teshale, Yigizie Yeshaw, Adugnaw Zeleke Alem.

**Software:** Hiwotie Getaneh Ayalew, Alemneh Mekuriaw Liyew, Zemenu Tadesse Tessema, Misganaw Gebrie Worku, Getayeneh Antehunegn Tesema, Tesfa Sewunet Alamneh, Achamyeleh Birhanu Teshale, Yigizie Yeshaw, Adugnaw Zeleke Alem.

**Supervision:** Hiwotie Getaneh Ayalew, Alemneh Mekuriaw Liyew, Zemenu Tadesse Tessema, Misganaw Gebrie Worku, Getayeneh Antehunegn Tesema, Tesfa Sewunet Alamneh, Achamyeleh Birhanu Teshale, Yigizie Yeshaw, Adugnaw Zeleke Alem.

**Validation:** Hiwotie Getaneh Ayalew, Alemneh Mekuriaw Liyew, Zemenu Tadesse Tessema, Misganaw Gebrie Worku, Getayeneh Antehunegn Tesema, Tesfa Sewunet Alamneh, Achamyeleh Birhanu Teshale, Yigizie Yeshaw, Adugnaw Zeleke Alem.

**Visualization:** Hiwotie Getaneh Ayalew, Alemneh Mekuriaw Liyew, Zemenu Tadesse Tessema, Misganaw Gebrie Worku, Getayeneh Antehunegn Tesema, Tesfa Sewunet Alamneh, Achamyeleh Birhanu Teshale, Yigizie Yeshaw, Adugnaw Zeleke Alem.

**Writing – original draft:** Hiwotie Getaneh Ayalew, Alemneh Mekuriaw Liyew, Zemenu Tadesse Tessema, Misganaw Gebrie Worku, Getayeneh Antehunegn Tesema, Tesfa Sewunet Alamneh, Achamyeleh Birhanu Teshale, Yigizie Yeshaw, Adugnaw Zeleke Alem.

**Writing – review & editing:** Hiwotie Getaneh Ayalew, Alemneh Mekuriaw Liyew, Zemenu Tadesse Tessema, Misganaw Gebrie Worku, Getayeneh Antehunegn Tesema, Tesfa Sewunet Alamneh, Achamyeleh Birhanu Teshale, Yigizie Yeshaw, Adugnaw Zeleke Alem.

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
