## [Decision Letter · Decision Letter 0]

4 Apr 2022

PONE-D-21-33331

SPATIAL VARIATION AND FACTORS ASSOCIATED WITH HOME DELIVERY AFTER ANC VISIT IN ETHIOPIA; SPATIAL AND MULTILEVEL ANALYSIS

PLOS ONE

Dear Dr. Hiwotie Getaneh,

Thank you for submitting your manuscript to PLOS ONE. After careful consideration, we feel that it has merit but does not fully meet PLOS ONE’s publication criteria as it currently stands. Therefore, we invite you to submit a revised version of the manuscript that addresses the points raised during the review process.

Kindly note that the reviewers have raised a number of concerns regarding methodology / analysis and also on the potential new insights from the study. The revised version should adequately address these concerns.

We look forward to receiving your revised manuscript.

Kind regards,

Prafulla Kumar Swain, Ph.D.

Academic Editor

PLOS ONE

https://journals.plos.org/plosone/s/file?id=ba62/PLOSOne_formatting_sample_title_authors_affiliations.pdf".

“All authors declare that they have no competing interests.”

3. Thank you for submitting the above manuscript to PLOS ONE. During our internal evaluation of the manuscript, we found significant text overlap between your submission and the following previously published work.

-https://www.researchsquare.com/article/rs-55326/v1

Please revise the manuscript to rephrase the duplicated text, cite your sources, and provide details as to how the current manuscript advances on previous work. Please note that further consideration is dependent on the submission of a manuscript that addresses these concerns about the overlap in text with published work.

Reviewers' comments:

Reviewer's Responses to Questions

**Comments to the Author**

1. Is the manuscript technically sound, and do the data support the conclusions?

Reviewer #1: Yes

Reviewer #2: Yes

2. Has the statistical analysis been performed appropriately and rigorously? 

Reviewer #1: Yes

Reviewer #2: No

3. Have the authors made all data underlying the findings in their manuscript fully available?

Reviewer #1: No

Reviewer #2: Yes

4. Is the manuscript presented in an intelligible fashion and written in standard English?

Reviewer #1: No

Reviewer #2: No

5. Review Comments to the Author

Reviewer #1: I am happy and glad to the editor for sending me the paper for reviewing.

I am happy reading the version where authors addressed an important topic, however I have some observations.

Please re-edit the conclusion of abstract. Is is liking odd to read. It is better.. starting of two sentences. It is large and the repetition of the conclusion of the main manuscript. I wonder authors are working with Ethiopia data but in the conclusion they wrote “It is better to give special attention for women who lived in Somalia, Amhara, Afar, Oromia, Benishangul Gumuz, Tigray and SNNP regions” do you have any clarification for this?

In the introduction section I didn’t found the strong rationality of the study, that is completely missing.

I would suggest to re-work on this to establish a strong rationality for the work.

In the first line of the introduction they told “skilled health care providers” I am fear for this line. Would it be “skilled birth attendants” please recheck.

“Studies have shown that mothers who have ANC visit to be more likely to give birth at health institutions” references missing for this line.

I think the Study setting is irrelevant, If I were you I would delete this section.

Authors used in some line “unskilled personal” or “unskilled personnel” please solve this issue.

“being in the poor” or “being poor”.

First two lines of the discussion is unnecessary.

“Possibly it might be most poor women in Ethiopia live in rural areas, in the absence of transportation and long-distance travel may lead to home birth.” References missing for this line.

Need “compared to urban women This finding..” full stop or anything else.

“This may also be due to distance to the health facilities, ambulance delays, inaccessibility of information, and low educational status of women from rural areas” does it means rural areas of Ethiopia are underprivileged, please extend this line for further clarification.

“The possible explanation might be due to scarcity of resources and poor clinical settings, in which most of the services are concentrated in urban areas or city administrates“ references missing.

In addition, those most agrarian women’s are farmers mostly had sudden labor onset and short duration which… I really fear of this line. Do you have any evidence or supporting literature.

Reviewer #2: The present paper has many shortcomings that are listed below:-

1. Why the marital status (married/unmarried) variable included in individual level variables.

2. Why wealth status is coded in 3 categories? The DHS data has 5 categories.

3. Media exposure can be coded as no, partial, full.

4. What does it mean birth interval (short/long)?

5. Why both the residency and region variables included in the models? It could lead to the problem of multicollinearity.

6. Home delivery after ANC visit was spatially clustered in Ethiopia. What is the role of independent variables in this phenomena?

7. The manuscript has many grammatical and typing errors.

6. PLOS authors have the option to publish the peer review history of their article (what does this mean?). If published, this will include your full peer review and any attached files.

Reviewer #1: No

Reviewer #2: No

---

## [Author Response · Author response to Decision Letter 0]

7 May 2022

Rebuttal letter Date April 20,2022

Subject; submission of revised manuscript (PONE-D-21-33331)

SPATIAL VARIATION AND FACTORS ASSOCIATED WITH HOME DELIVERY AFTER ANC VISIT IN ETHIOPIA; SPATIAL AND MULTILEVEL ANALYSIS

Hiwotie Getaneh Ayalew

To PLOS ONE

Dear all,

We would like to thank you for these constructive, building, and improvable comments on this manuscript that would improve the substance and content of the manuscript. We considered each comment and clarification question of editors and reviewers on the manuscript thoroughly. Our point-by-point responses for each comment and question are described in detail on the following pages. Further, the details of changes were shown by track changes in the supplementary document attached. The manuscript language was further improved in the revised manuscript. and we follow journal guidelines. I have attached recent comments in a point-by-point response.

Version 1; editor’s comments

Authors’ response; thank you dear editor we have prepared the documents based on PLOS ONE requirements.

2. Thank you for stating the following in your Competing Interests section. 

Authors’ response; thanks, dear editor. 

3. Thank you for submitting the above manuscript to PLOS ONE. During our internal evaluation of the manuscript, we found significant text overlap between your submission and the following previously published work.

-https://www.researchsquare.com/article/rs-55326/v1

Authors’ response; Thank you, dear editor, we have included the reference and we have rephrased the duplicated texts and words in the revised manuscript.

Version 2; reviewers’ comments

1. Is the manuscript technically sound, and do the data support the conclusions?

Reviewer #1: Yes

Reviewer #2: Yes

Authors’ response; thank you, dear reviewers

2. Has the statistical analysis been performed appropriately and rigorously?

Reviewer #1: Yes

Reviewer #2: No

Authors’ response; thank you, dear reviewers. We have improved the revised version of the manuscript.

3. Have the authors made all data underlying the findings in their manuscript fully available?

Reviewer #1: No

Reviewer #2: Yes

Authors’ response; Thank you, dear reviewers. We have stated that the data was fully available without restriction in the revised manuscript at the declaration session.

4. Is the manuscript presented in an intelligible fashion and written in standard English?

Reviewer #1: No

Reviewer #2: No

Authors’ response; thanks a lot dear reviewer. we have critically improved the readability of the manuscript. Please see the revised version.

Reviewer 1 Comments 

After having carefully read the manuscript and taking notes on overall strengths and weaknesses, I share my comments below.

General comments and suggestions 1; I am happy and glad to the editor for sending me the paper for reviewing.I am happy reading the version where authors addressed an important topic, however I have some observations. Please re-edit the conclusion of abstract. Is is liking odd to read. It is better.. starting of two sentences. It is large and the repetition of the conclusion of the main manuscript. I wonder authors are working with Ethiopia data but in the conclusion they wrote “It is better to give special attention for women who lived in Somalia, Amhara, Afar, Oromia, Benishangul Gumuz, Tigray and SNNP regions” do you have any clarification for this?

Authors’ response; thanks a lot dear reviewer for your critical comment to improve the manuscript. we have fully accepted your comment we have to avoid and reduce the repetition of the conclusion of the abstract.

These regions Somalia, Amhara, Afar, Oromia, Benishangul Gumuz, Tigray and SNNP regions were detected when we conduct spatial analysis. we find hotspot regions in hot spot analysis and high-risk areas of home delivery after ANC visit with spatial scan statistics scanning window in these regions. Therefore based on the finding it is important to recommend those regions. We kindly request to see the revised manuscript at the spatial analysis result session.

Comment 2; In the introduction section I didn’t found the strong rationality of the study, that is completely missing. I would suggest to re-work on this to establish a strong rationality for the work.

Authors’ response; thanks a lot dear reviewer for your unlimited effort to improve the manuscript. We accepted your comment and included sound justification in the revised manuscript. 

Comment 3; In the first line of the introduction they told “skilled health care providers” I am fear for this line. Would it be “skilled birth attendants” please recheck.

Authors’ response; thanks, dear reviewer for your detailed revision. We have accepted the comment and we correct the revised manuscript.

Comment 4; Studies have shown that mothers who have ANC visits are more likely to give birth at health institutions” references missing from this line.

Authors’ response; Thanks a lot dear reviewer. We have added the reference in the revised manuscript.

Comment 5; I think the Study setting is irrelevant, If I were you I would delete this section. Authors used in some line “unskilled personal” or “unskilled personnel” please solve this issue.

Authors’ response; thank you in advance dear reviewer. We have delated the study setting and we also have taken an edition on the issue of “unskilled personal” in the revised manuscript. Comment 6; “being in the poor” or “being poor”. First two lines of the discussion is unnecessary. 

Authors’ response; thank you dear reviewer we have written the interpretation of the variable wealth index as “being in the poor” and we have also removed the first line of the discussion session in the revised manuscript.

Comment 7; “Possibly it might be most poor women in Ethiopia live in rural areas, in the absence of transportation and long-distance travel may lead to home birth.” References missing for this line

Authors’ response; thanks dear reviewer we have to appreciate your comments. We have added some references and we kindly request you to see them in the revised manuscript.

Comment 8; Need “compared to urban women This finding.” full stop or anything else.

“This may also be due to distance to the health facilities, ambulance delays, inaccessibility of information, and low educational status of women from rural areas” does it means rural areas of Ethiopia are underprivileged, please extend this line for further clarification.

Authors’ response; thanks a lot dear reviewer for your deep comments. We have added a full stop at the end of “compared to urban women” Additionally for the next comment we have further extended our clarification for the explanation. Since it does not mean that rural women are underprivileged rather it means due to the topographic difficulty of the country and scattered living distribution of households and health institutions, there is long distance and difficult roads so this leads to home delivery. We also added a reference for this comment in the revised manuscript.

Comment 9; The possible explanation might be due to scarcity of resources and poor clinical settings, in which most of the services are concentrated in urban areas or city administrates “references missing.

Authors’ response; thank you, dear reviewer. We have added a reference and we kindly asked you to see the revised manuscript.

Comment10; In addition, those most agrarian women’s are farmers mostly had sudden labor onset and short duration which… I really fear of this line. Do you have any evidence or supporting literature?

Authors’ response; thanks dear reviewer for your constructive comments. We have added evidence of supporting kinds of literature in the revised manuscript. We kindly ask you to see the revised manuscript.

Reviewer #2 comments

Comment 1. Why the marital status (married/unmarried) variable included in individual level variables.

Authors’ response; thanks dear reviewer we accepted your comment. The analytic model of this study was the multilevel model and it includes individual and community level factors.so factors that can be directly found in the data set for each study participants were considered as individual-level factors like marital status. Whereas community-level factors were generated by aggregating individual-level factors.

Comment 2; why wealth status is coded in 3 categories? The DHS data has 5 categories.

Authors’ response; thank your dear reviewer for your comment. DHS has five categories for wealth index, but for our analysis, we have recoded wealth index into three categories, due to that it did not fulfill the chi-square assumption as we used it as five categories so we changed it to three categories. Some kinds of literature also use wealth index as three categories for analysis

comment 3; Media exposure can be coded as no, partial, full

Authors’ response; thank you, dear reviewer. we have used media exposure for analysis as exposed (yes)and unexposed (no) for media.

. Comment 4; what does it mean birth interval (short/long)? 

Authors’ response; thank you, dear reviewer. We strongly appreciate your comment. Birth interval is the duration which refers to the time interval from delivery to the next subsequent pregnancy of the fetus. It is classified as short if it is less than 15 months and longer if it is greater than or equal to 15 months and it also has an effect on place of delivery as delivered in Textbook of Williams Obstetrics 25th edition.

 Comment 5; why both the residency and region variables included in the models? It could lead to the problem of multicollinearity. 

Authors’ response; Thanks a lot dear reviewer for your interesting and constructive comments. We initially checked the VIF (variance inflation factor) of all individual and community level variables before fitting the model and there was no multicollinearity in between variables.

comment 6; Home delivery after ANC visit was spatially clustered in Ethiopia. What is the role of independent variables in this phenomenon? 

Authors’ response; thanks dear reviewer we appreciate your comments. In this study, we used two models spatial and multilevel analysis. In the case of spatial analysis, we only used the GPS data set and the outcome variable to identify high-risk areas of home delivery after ANC visit in Ethiopia, whereas in the multilevel model to identify determinants of home delivery after ANC visit we used independent variables (individual level and community level factors). therefore independent variables for spatial analysis to determine whether home delivery was spatially clustered or not have no role.

Comment 7; the manuscript has many grammatical and typing errors. 

Authors’ response; thanks a lot dear reviewer we have critically improved the grammatical error and readability of the manuscript. Please see the revised version.

Thanks, a lot!!!

---

## [Decision Letter · Decision Letter 1]

28 Jul 2022

SPATIAL VARIATION AND FACTORS ASSOCIATED WITH HOME DELIVERY AFTER ANC VISIT IN ETHIOPIA; SPATIAL AND MULTILEVEL ANALYSIS

PONE-D-21-33331R1

Dear Dr. Hiwotie Getaneh Ayalew,

We’re pleased to inform you that your manuscript has been judged scientifically suitable for publication and will be formally accepted for publication once it meets all outstanding technical requirements.

Kind regards,

Prafulla Kumar Swain, Ph.D.

Academic Editor

PLOS ONE

Additional Editor Comments (optional):

Authors have addressed the concern raised by the reviewers, now the revised version of the manuscript may be accepted for publication.

Reviewers' comments:

Reviewer's Responses to Questions

**Comments to the Author**

1. If the authors have adequately addressed your comments raised in a previous round of review and you feel that this manuscript is now acceptable for publication, you may indicate that here to bypass the “Comments to the Author” section, enter your conflict of interest statement in the “Confidential to Editor” section, and submit your "Accept" recommendation.

Reviewer #1: All comments have been addressed

Reviewer #2: All comments have been addressed

2. Is the manuscript technically sound, and do the data support the conclusions?

Reviewer #1: Yes

Reviewer #2: Yes

3. Has the statistical analysis been performed appropriately and rigorously? 

Reviewer #1: Yes

Reviewer #2: Yes

4. Have the authors made all data underlying the findings in their manuscript fully available?

Reviewer #1: Yes

Reviewer #2: Yes

5. Is the manuscript presented in an intelligible fashion and written in standard English?

Reviewer #1: Yes

Reviewer #2: Yes

6. Review Comments to the Author

Reviewer #1: Thanks to the all authors addressing my previous comments, I am happy reading this version, I don’t have further comments. Congratulations to all.

Reviewer #2: All the comments have been addressed and MS looks greater now. In its current form, MS may be published.

7. PLOS authors have the option to publish the peer review history of their article (what does this mean?). If published, this will include your full peer review and any attached files.

Reviewer #1: No

Reviewer #2: No

---

## [Editor Report · Acceptance letter]

16 Aug 2022

PONE-D-21-33331R1 

SPATIAL VARIATION AND FACTORS ASSOCIATED WITH HOME DELIVERY AFTER ANC VISIT IN ETHIOPIA; SPATIAL AND MULTILEVEL ANALYSIS 

Dear Dr. Ayalew:

I'm pleased to inform you that your manuscript has been deemed suitable for publication in PLOS ONE. Congratulations! Your manuscript is now with our production department. 

Kind regards, 

on behalf of

Dr. Prafulla Kumar Swain 

Academic Editor

PLOS ONE